**Data Availability Statement:** All files are available at https://doi.org/10.32873/unl.dr.20210928.

**Funding:** Partial funding for this research was received from a grant jointly provided by the U.S.

# Emotional responses to COVID-19 stressors increase information avoidance about an important unrelated health threat

Christopher R. Gustafson[1]*, Kathleen R. Brooks[1], Syed Imran Ali Meerza[2], Amalia Yiannaka[1]

1 Department of Agricultural Economics, University of Nebraska-Lincoln, Lincoln, Nebraska, United States of America, 2 Department of Agriculture, Arkansas Tech University, Russellville, Arkansas, United States of America

⊙ These authors contributed equally to this work.
* cgustafson6@unl.edu

## Abstract

The COVID-19 pandemic, like other crises, has had direct and indirect impacts on individuals, many of which have been negative. While a large body of research has examined the impacts of COVID-19 on people's lives, there is little evidence about how COVID-19 affects decision-making broadly. Emotional responses to COVID-19-related stressors, such as illness and income loss, provide a pathway for these stressors to affect decision-making. In this study, we examine linkages between exposure to COVID-19-related stressors—focusing on temporally specific local case counts and loss of income due to the pandemic—and decisions to access information about antimicrobial resistance (AMR), another critically important health issue. COVID-19 constitutes a natural experiment in that people's exposure to stressors does not result from those individuals' current decisions. Using a nationally representative survey with 1223 respondents in December 2020, we linked the temporally specific COVID-19 cases and income loss experienced by participants to an increased likelihood of feeling hopeless. Higher feelings of hopelessness led to a higher probability of avoiding information about AMR. A mediation analysis confirms that emotional responses to COVID-19 stressors significantly increase information avoidance about an unrelated, but important health issue. Our results suggest that large-scale crises, like COVID-19 and climate change, may diminish action on other important health issues facing humanity.

## 1. Introduction

Large-scale crises—e.g., conflicts, pandemics, and natural disasters—decrease individuals' well-being directly but can also have secondary effects by impacting decision-making at individual and societal levels. Documented secondary effects of the COVID-19 pandemic include decreased access to education [1], delayed or foregone medical care [2] that may contribute to additional deaths [3], and loss of income through furloughs or layoffs [4], among others.

While some impacts—such as school closures or delaying elective surgeries—were decisions made by organizations or governments to limit the spread of COVID-19, individual

Meat Animal Research Center (USMARC) of the U. S. Department of Agriculture (USDA) and the University of Nebraska-Lincoln Agricultural Research Division (UNL ARD); grant number 3904.

**Competing interests:** The authors have declared that no competing interests exist.

decision-making can also change in response to pandemic-related stressors, potentially impacting health and well-being. A well-documented impact of COVID-19–as well as other crises—is an increase in negative emotions, such as hopelessness. Numerous articles document high levels of hopelessness in healthcare workers and patients during the COVID-19 pandemic [5–9], and predictors of hopelessness in geographically diverse samples [10, 11]. Researchers have also examined pathways through which COVID-19 might negatively impact emotions, such as through threats to health, financial well-being, changes in routine, and via increased social isolation. Findings suggest that economic, health, and routine changes had impacts on emotion [12–14]. In a study on the impact of COVID-related stressors in the US, Luk et al. [15] documented that nearly 50% of respondents experienced significant stress, which corresponded to poorer health outcomes, while a study of adults in Italy found similarly high levels of mental health disorders [16]. Both studies found that job and income-related stressors predicted higher levels of emotional impact.

A few studies have compared mental health outcomes in populations before and during COVID-19. A study of US adults found that symptoms of serious psychological distress had nearly quadrupled in 2020 relative to levels found in 2018 [17], while a 2020 study of adults in Wales showed clinically significant psychological distress in approximately 50%, and severe distress in 20%, of respondents [18]. These levels were 3–4 times higher than in 2019 surveys.

National-level data from US Census Bureau and Centers for Disease Control and Prevention (CDC) surveys show a marked increase in general levels of hopelessness among adult residents of the US during the pandemic relative to pre-pandemic levels. Approximately 50 percent of respondents to multiple waves of a US Census Bureau survey (distributed April 2020-July 2021) experienced feeling hopeless, down, or depressed in the seven days preceding the survey throughout 2020–21, while only 17.6 percent had reported those feelings in 2019 [19, 20].

Numerous studies document the effect of emotion on decision-making. Certain negative emotions, such as sadness, have been found to increase the perceived probability of an adverse event occurring [21–23], and lead to more risk-averse decisions [24]. A significant increase in the delay or avoidance of medical care has been documented during the COVID-19 pandemic [2], placing individuals at risk for more serious illness or mortality. Negative emotions may have led individuals to delay or avoid medical appointments by increasing the perceived risk of contracting COVID-19 at the healthcare facility. Decisions to delay or avoid medical care during the early stages of the COVID-19 pandemic have had real consequences. A "crisis of undiagnosed cancers" has been reported [25], with estimates that 10,000 additional deaths from colorectal and breast cancers will occur by 2030 [26].

Recently, a behavioral phenomenon, information avoidance, has gained increasing attention because it has important implications for individual behaviors related to health. Researchers have found that many individuals will intentionally avoid accessing freely available information, including information that could influence the healthiness of eating behaviors and decisions to access test results to learn about one's disease state [27, 28]. Information avoidance may affect financial decision-making; personal relationships; environmental conservation; and "license" to take self-interested actions [29–31]. Information avoidance behaviors have been widely documented in an array of health-related settings, including choosing to view information about nutrition [27]; receiving tests for diseases such as cancer, diabetes, and HIV [28, 32–36]; and learning about important challenges facing the human health sector, such as antimicrobial resistance (AMR) [37].

Emotion was identified as a driver of information avoidance early in the literature [38]. Anticipated negative emotional reactions were listed as one of the three primary reasons that information may be avoided [39]. Hope, as well as feelings related to hope, has specifically

been highlighted as a driver of decisions to avoid or access information. Hope is defined as the perception that one has the ability to identify and use pathways to achieve desired outcomes [40]. People are markedly more likely to avoid exposure to information with potentially negative implications when they lack hope—that is, when they feel hopeless—about being able to address the issue [37, 41–43]. While not explicitly framing the issue in terms of hopelessness, information avoidance has been found to be common for threats perceived to be unmanageable [37, 41–43]. On the other hand, believing that one can mitigate negative information that may be communicated to them decreases avoidance of information [44–46]. Possessing the capacity and mental health resources to manage threats—by feeling optimistic, employing emotional coping strategies, or perceiving the agency to act, among others—decreases information avoidance, while feeling hopeless or powerless to do anything about a situation increases avoidance behavior across both health and non-health domains [37, 43, 47–50].

AMR is a societally important health challenge that affects approximately three million individuals and causes nearly 50,000 deaths annually in the US [51]. Worldwide, AMR is estimated to cause 700,000 deaths annually; this number is predicted to increase to 10 million per year by 2050 [51]. Information that can facilitate positive individual or collective actions to address AMR, as well as other important social challenges, is readily available [52–54]. Information shapes beliefs and affects individual behavior, like product purchases in markets, and collective behavior, like voting [55–58]. In the case of AMR, private actions might include purchasing foods from animals raised without the use of unnecessary antibiotics to incentivize producers to implement judicious use of antibiotics. Increasing investment in developing new antibiotics and other antimicrobials—which could be facilitated with public support for research and development—has been highlighted as a critical need to prevent further increases in illnesses and deaths due to AMR [59]. Avoiding information about AMR will likely suppress private actions through markets and support for public policies to address low levels of investment in developing new antimicrobials.

In this study, we examine a novel question: can one crisis—the COVID-19 pandemic—affect the decision to access or avoid information in unrelated but critical personal and societal health issues by affecting general feelings of hopelessness? We use exogenous variation in exposure to COVID-19-related stressors—income loss and local COVID-19 cases, which have been identified as the COVID-19-related outcomes that most impact mental health [12]—to examine the impact of a large-scale crisis on decisions to avoid information about AMR. Both income loss and COVID-19 cases are easily measurable and have important, widely discussed policy implications related to mask mandates and government spending programs, for instance. However, if COVID-19 impacts behavior in other domains—such as by increasing avoidance of AMR information—addressing the stressors that promote these spillover effects will be even more important. An important contribution of this study is to understand whether COVID-19 imposes costs in other health domains by reducing action on AMR.

While the impact of COVID-19 on medical care has been reported in other studies [2, 25], we examine decisions to access/avoid AMR information because AMR poses an important threat to human health, like COVID-19, and should therefore be highly salient to respondents, but the decision to access AMR information may also be influenced by emotions like hopelessness. Due to the COVID-19 pandemic, health concerns have likely had greater cognitive accessibility—how readily an individual thinks about a topic [60], which should make individuals less likely to avoid information [61]. However, while both COVID-19 and AMR pose significant threats to human health, they involve different pathogens and transmission routes. Thus, accessing information about how to reduce the threat of AMR is still critical because many of those actions differ from actions needed to prevent the risk of COVID-19.

We estimate the effect of exogenous, objective COVID-19 measures—average new county-level COVID-19 cases over the two weeks prior to survey completion and loss of income during the COVID-19 pandemic—on hopelessness experienced during the previous week. These variables—COVID-19 cases and income loss—represent health and economic concerns, which have been identified as having the greatest effect on mental health during the COVID-19 pandemic [12]. Next, we evaluate the impact of hopelessness on AMR information avoidance using our survey data. Finally, we conduct a mediation analysis to examine the direct effect of COVID-19 stressors on AMR information avoidance, as well as an indirect pathway from stressors to emotional response to information avoidance.

## 2. Materials and methods

### 2.1. Survey design and data

We developed a survey about consumer perceptions and knowledge of antibiotic use and resistance to incorporate questions about the respondents' emotional state in the previous seven days as well as questions about COVID-19-related economic impacts leading to decreased household income, such as decreased earnings or loss of a job. The custom AMR survey was distributed by IRi (www.iriworldwide.com) to adults ≥19 years of age in a consumer panel designed to be representative of the US population according to key demographic variables, receiving 1223 completed surveys. IRi, a leading survey firm, uses a probability-based consumer panel designed to be representative of the US population for key demographic variables. The research was approved by the authors' university ethical review board (protocol #20180418265EX). Participants provided written informed consent. No data were collected from minors.

To generate a measure of information avoidance in the context of AMR, respondents were given a choice to watch a video produced by the Food and Agriculture Organization of the United Nations (FAO) about antimicrobial resistance, entitled "Antimicrobial resistance: the role of food and agriculture," (available at https://www.youtube.com/watch?v=d3YXW_gWNz4) or an unrelated video of identical length featuring a black screen and providing no information ("Nature white noise: rain and thunderstorm sounds for relaxation"). The video, "The Role of Food and Agriculture in AMR", discusses the risk posed by antimicrobial use in livestock systems to the development of antibiotic-resistant bacteria. The choice to watch the "Nature white noise" video was categorized as information avoidance behavior.

Extending previous literature on information avoidance, we focus on hopelessness as a pathway through which one crisis might engender information avoidance behavior in another domain. We adopted questions used by the Pew Research Center in surveys they administered during the COVID-19 pandemic for comparability to other data [62]. The Pew survey asked about feeling hopeful about the future in four categories, ranging from "Rarely or none of the time" to "Most or all of the time;" there was additionally a "prefer not to answer" option. Participants in our survey noted the number of days in the week that preceded taking the survey that they felt hopeful about the future. To link this question more directly with comparable U.S. Census data from the Household Pulse Survey (HPS), we recoded the variable to represent hopelessness. To convert responses to this question into "hopelessness," we recoded the four categories so that a respondent who felt hopeful "Most or all of the time" was coded as feeling hopeless "rarely or none of the time," while an individual who felt hopeful rarely or none of the time was recoded as feeling hopeless most or all of the time, and so on. We include hopelessness as a categorical variable in our analyses.

We also used data about income loss in the respondents' households resulting from COVID-19 as another exogenous variable that has been found to cause a strong negative

emotional impact [12], which can lead to feelings of hopelessness. Participants responded to two questions that asked about losses in income. Participants first noted whether anyone in the household had been laid off or lost a job because of the COVID-19 pandemic, while in the second, they responded to a question about taking a pay cut due to reduced hours or reduced demand for their work. The respondent could indicate that it had already happened, that it had not yet happened, but they expected it to in the future, or that it had not happened, and they did not expect it to. To generate a variable capturing loss of income, we combined responses to the questions about losing a job and receiving a pay cut/reduced hours to create one variable indicating whether the household had 1) already experienced income loss from either job loss or reduced hours or pay, 2) had not yet experienced income loss but expected to (again through either route), or 3) had not experienced income loss and did not expect to.

Additional survey questions included demographic information, including the gender, age, and number of years of education of the respondent, and household income. We processed these variables for use as control variables in our analysis. A new variable, Female, was created from responses to the question about gender, which took the value 1 if the respondent indicated that they were female and the value 0 otherwise. We collected data on age and household income in ranges. We used the midpoint of the range to convert these into numeric variables. Finally, data on education were collected in terms of the highest level of education completed. We converted this information into a numeric variable by taking the expected amount of time to complete a particular degree: a high school diploma/GED was assigned a value of 12 years of education, while a bachelor's degree was coded as 16 years of education, for instance.

To link data on hopelessness collected in the survey to local, time-specific COVID-19 conditions, respondents reported their zip code. We used data on zip codes to link the participant's survey responses to temporally matched county-level information about the COVID-19 pandemic when the individual responded to the survey and to information about the population of the county. We downloaded county and date-specific COVID-19 case information from the New York Times, which were compiled from reports from state and local health agencies [63]. For zip codes that cross county lines, we averaged COVID-19 case data from the counties that fell within the zip code to create one measure for that respondent. We created a normalized measure of the number of cases in the respondents' location—cases per 100,000 individuals—using data on COVID-19 cases and county population. We calculated the average number of cases per 100,000 individuals over the two weeks prior to the date the respondent submitted their survey response to capture the severity of the pandemic in the time period in which the respondent completed the survey.

## 2.2. Analysis

Our analyses link the evidence for COVID-19-related exogenous stressors with information avoidance in the context of AMR. We focus on hopelessness as a mediator variable that ties COVID-19 stressors to AMR information avoidance, as it has been identified as a driver of information avoidance in other contexts. We study two exogenous COVID-19 outcomes related to what have been identified as the greatest COVID-19-related threats to mental health: health and economic threats [12]. Our outcomes, cases in the local area and loss of income, are both easily measurable and have important, widely discussed policy implications related to mask mandates and government spending programs, for instance. An important contribution of this study is to understand whether COVID-19 imposes costs in other domains by reducing action on antimicrobial resistance.

First, we tested for evidence of the relationship of COVID-19 cases/100,000 residents and income loss to hopelessness. We estimated this relationship using an ordered logistic

regression and conducted two versions of the regression. In the first version, COVID-19 cases and income loss were the sole independent variables. Then, we added common demographic characteristics—gender, age, education, and household income—to check for robustness of the primary results.

Next, we examined the relationship of hopelessness to AMR information avoidance using binary logistic regression models. The outcome variable documenting AMR information avoidance was the choice to access or avoid a video about AMR in food and agriculture, which is a binary variable (= 1 if the respondent chose to avoid the video about food, agriculture, and AMR; and = 0 if they chose to access the AMR video). A second version added local cases of COVID-19 and income loss in addition to hopelessness to identify whether these COVID-19-related stressors directly influence AMR information avoidance behavior or only act indirectly through their effect on hopelessness. Next, we conducted a version of the regression that incorporated demographic variables. Finally, we examined AMR information avoidance behavior with hopelessness, COVID-19 stressors, and demographic characteristics as independent variables.

To examine whether hopelessness serves as a significant pathway through which COVID-related stressors impact decisions to access or avoid information about AMR, we conducted a mediation analysis of the impact of the economic COVID-19-related stressor on AMR information avoidance. Mediation analysis uses bootstrapping techniques to estimate the significance of the impact of an exogenous variable (the COVID-19 stressor related to experienced or expected loss of income in this case) acting directly on AMR information avoidance, or indirectly through a variable mediating the relationship—hopelessness, here—between the exogenous stressor variable and AMR information avoidance [64]. We reported results that are significant with p-values $< 0.05$.

## 3. Results

Avoidance behavior was common in the AMR survey. Nearly 40% of AMR survey respondents chose not to view the AMR video. Data comparing characteristics of respondents of the custom AMR survey with the US population are presented in Table 1. The custom IRi sample is representative of the adult US population by age, income, and sex, but IRi respondents have higher levels of education on average than the overall adult U.S. population.

We compared the two stressors through which the pandemic might impact people—income loss and new local COVID-19 cases—between AMR data and data from week 21—the same time period that AMR data were collected—of the US Census Bureau's Household Pulse Survey (HPS), which has been administered throughout the pandemic. Just over 30% of AMR respondents had experienced a loss of income during the COVID-19 pandemic, with an additional 15.5% of AMR respondents expected/worried about a loss of income in the future due

**Table 1. Summary statistics.**

|  | AMR Survey | US Population |
|---|---|---|
| Female (%) | 53% | 51% |
| Age (% Adults < 65) | 80% | 79% |
| Education (% ≥ Bachelor's Degree) | 51% | 32% |
| Household income (Median) | $62,500 | $62,843 |

Notes: Data are from 1223 responses to AMR Survey from custom survey (Dec. 16–31, 2020); US population data from US Census Bureau Quick Facts [65]. Estimates for income omit data from respondents who did not report income.

**Table 2. Distribution of responses to questions about loss of employment income.**

| | Experienced loss of employment income | Expected loss of employment income | No loss of employment income |
|---|---|---|---|
| AMR survey (N = 1223) | 30.1% | 15.5% | 54.4% |
| Household Pulse Survey, week 21 (HPS21) (N = 59,644) | 40.4% | 23.4% | 36.2% |

Notes: Questions in the AMR survey and the Household Pulse Survey were implemented differently. While respondents to the Household Pulse Survey responded separately to questions about experienced loss of employment income and expected loss of employment income in the next four weeks, our survey asked participants to select one of four mutually exclusive responses: 1) they have already experienced a loss of income; 2) they have not yet but worry that they will experience a loss of income (no time frame); 3) they have not and do not expect to; 4) prefer not to answer. The percentages in the tables were calculated omitting responses from individuals who preferred not to answer (AMR data) or who did not respond to the questions (HPS data).

to COVID-19. Approximately 40% of respondents to the week 21 HPS had experienced a loss of income between March 13, 2020 and the survey period (December 9–21, 2020) while 23.4% of HPS respondents expected a loss of employment income in the next four weeks (Table 2).

Average new COVID-19 cases over the two weeks prior to the survey were quite similar in the AMR and national data (Table 3). Median cases per 100,000 residents were 59.5 in the AMR sample, while in the US overall there were 59.1 cases/100,000 during the same time period. Mean cases/100,000 in the AMR sample were 63.1; in the US, there were 65.9 cases per 100,000 residents.

We additionally used HPS data on hopelessness to compare our AMR survey with HPS21 data (a period overlapping with our survey: Dec. 9–21, 2020), and a pre-pandemic CDC National Health Interview Survey (NHIS) from 2019 (Table 4) [19, 20]. Similar percentages of AMR and HPS21 respondents who answered the survey question about feeling hopeless (65.7% of the AMR sample and 67.6% of the HPS21 sample) reported feeling some level of hopelessness during the previous week. The percentage of respondents to the pre-pandemic 2019 NHIS survey experiencing levels of hopelessness (17.6% total across hopelessness categories) was markedly lower than in AMR or HPS datasets. One-third of the AMR sample and only 25% of the HPS21 sample experienced little to no hopelessness during this period. In contrast, hopelessness was rare in the pre-pandemic NHIS survey—82% were hopeless rarely/none of the time.

### 3.1 The impact of COVID-19-related stressors on hopelessness

We found that experienced loss of income by the respondents' households and the number of cases in the respondents' counties significantly increased the likelihood of experiencing higher levels of hopelessness in the AMR data (Table 5). Each additional daily case of COVID-19 per 100,000 residents over the previous two weeks increased the likelihood of experiencing a higher level of hopelessness by a small, but statistically significant amount (adjusted odds ratio (aOR) = 1.005, 95% CI: 1.000, 1.009, $p = 0.03$). Respondents who had experienced a loss of income were 1.5 times (95% CI: 1.158, 1.977, $p = 0.002$) more likely to experience higher levels

**Table 3. COVID-19 cases per 100,000 residents in respondents' county (AMR) and in all US counties.**

| | Median | Mean | Standard Deviation |
|---|---|---|---|
| AMR survey (Dec. 16–31, 2020) | 59.5 | 63.1 | 26.7 |
| All US Counties (Dec. 16–31, 2020) | 59.1 | 65.9 | 39.3 |

Data from New York Times COVID-19 monitoring project [63].

**Table 4. Distribution of responses to question about feelings of hopelessness.**

|  | Rarely or none of the time | Some or a little of the time | Occasionally or a moderate amount of the time | Most or all of the time | Prefer not to answer |
|---|---|---|---|---|---|
| AMR (N = 1223) | 32.5% | 26.7% | 17.6% | 17.9% | 5.2% |
| HPS21 (N = 59,644) | 24.5% | 20.8% | 13.1% | 17.3% | 24.3% |
| 2019 NHIS (N = 31,289) | 82.1% | 12.5% | 2.2% | 2.8% | 0.3% |

Notes: Data from custom AMR survey (Dec. 16–31, 2020); US Census Household Pulse Survey, Week 21 (Dec. 9–21, 2020); and 2019 National Health Interview Survey [19, 20].

of hopelessness than those who had not. Expected loss of income was not significantly associated with increased feelings of hopelessness. Estimates of the association of case and income loss variables remained significant with the inclusion of demographic variables. The only demographic variable significantly related to level of hopelessness in the AMR survey was education. The odds of experiencing hopelessness were 0.875 (95% CI: 0.810, 0.944, p<0.001) times lower per additional year of education.

## 3.2. The effect of hopelessness on avoidance behavior

Feelings of hopelessness in general—not specifically about the threat of AMR—predicted avoidance of AMR information (Table 6). We estimated four versions of a model of the relationship between AMR information avoidance and hopelessness to check the robustness of results to the inclusion of objective health and economic COVID-19 stressors as well as demographic characteristics. Nearly constant feelings of hopelessness increased the likelihood of AMR information avoidance by 1.66 (95% CI: 1.11, 2.49, p = 0.01) to 1.95 (95% CI: 1.35, 2.82, p<0.001) times relative to individuals who did not experience feeling hopeless, while any feelings of hopelessness increased the likelihood of information avoidance by at least 1.49 times (95% CI: 1.05, 2.11, p = 0.02). These results are robust to the inclusion of the direct COVID-19 stressors—cases per 100K and income loss (Models II and IV)—as well as to the addition of demographic characteristics (Models III and IV). Among demographic variables, we found

**Table 5. Adjusted odds ratios (aOR) and 95% confidence intervals (95% CI) of the association of COVID-19 cases and income loss on feelings of hopelessness.**

|  | AMR Survey (Dec. 16–31, 2020) | |
|---|---|---|
|  | aOR (95% CI) | aOR (95% CI) |
| Cases/100K | 1.005* (1.000, 1.009) | 1.005* (1.000, 1.009) |
| Experienced income loss | 1.513** (1.158, 1.977) | 1.455** (1.095, 1.935) |
| Expected income loss | 1.243 (0.902, 1.712) | 1.153 (0.818, 1.624) |
| Female |  | 1.102 (0.879, 1.396) |
| Age |  | 0.997 (0.901, 1.104) |
| Education |  | 0.875*** (0.810, 0.944) |
| Income |  | 1.005 (0.999, 1.011) |
| AIC | 2529.70 | 2522.05 |

Notes: Data from AMR survey and case numbers compiled by the New York Times COVID-19 monitoring project at the state and county level.

*** = <0.001

** = <0.01

* = <0.05.

**Table 6. Adjusted odds ratios (aOR) and 95% confidence intervals (95% CI) of the association of feeling hopeless on antimicrobial resistance information avoidance with robustness checks for COVID-19 stressors and demographic characteristics.**

| | (I) aOR (95% CI) | (II) aOR (95% CI) | (III) aOR (95% CI) | (IV) aOR (95% CI) |
|---|---|---|---|---|
| Intercept | 0.42*** (0.34, 0.52) | 0.50*** (0.34, 0.73) | 0.53 (0.20, 1.37) | 0.63 (0.20, 2.00) |
| HOPELESS | | | | |
| All/ nearly all the time | 1.87*** (1.33, 2.64) | 1.95*** (1.35, 2.82) | 1.66* (1.11, 2.49) | 1.72** (1.14, 2.57) |
| Most of the time | 1.49* (1.05, 2.11) | 1.52* (1.05, 2.20) | 1.59* (1.06, 2.39) | 1.64* (1.09, 2.46) |
| Some of the time | 1.55** (1.14, 2.12) | 1.60** (1.15, 2.23) | 1.58* (1.11, 2.26) | 1.61** (1.13, 2.31) |
| Prefer not to answer | 1.62 (0.94, 2.78) | 1.40 (0.76, 2.56) | - | - |
| CASES/100K | | 1.00 (0.99, 1.00) | | 1.00 (0.99, 1.00) |
| INCOME LOSS | | | | |
| Experienced | | 0.80 (0.59, 1.07) | | 0.80 (0.57, 1.13) |
| Expected | | 1.04 (0.72, 1.49) | | 0.80 (0.52, 1.22) |
| Prefer not to answer | | 1.73 (0.94, 3.20) | | - |
| Female | | | 1.58** (1.19, 2.10) | 1.58** (1.19, 2.10) |
| Age | | | 1.00 (0.99, 1.01) | 1.00 (0.99, 1.00) |
| Education | | | 1.00 (0.93, 1.07) | 1.00 (0.93, 1.07) |
| Income | | | 1.00 (0.99, 1.00) | 1.00 (0.99, 1.00) |
| AIC | 1606.6 | 1436.2 | 1413.4 | 1159.4 |

Notes: Data from AMR survey. Models III and IV do not have estimates for "Prefer not to answer" for HOPELESS and INCOME LOSS because individuals submitting those responses also did not report household income; those observations were dropped in order to create a numeric income variable.

*** = <0.001

** = <0.01

* = <0.05.

that women were 1.58 times (95% CI: 1.19, 2.10, p = 0.009) more likely to avoid information about AMR than men, while no other demographic variables were statistically significant. Neither COVID-19 stressor was significantly related to information avoidance.

### 3.3 Mediation analysis: Income loss, hopelessness, and AMR information avoidance

Mediation analysis confirmed that hopelessness significantly mediated the impact of COVID stressors on AMR information avoidance (Table 7). The results showed that COVID-19-related stressors only increased the avoidance of information about AMR indirectly by increasing feelings of hopelessness. Mediation analysis reveals an average causal mediation effect ranging from 0.009 (95% CI = 0.001, 0.020, p-value = 0.04) when all demographic control variables are included to 0.015 (95% CI = 0.04, 0.30, p-value<0.001) when demographic variables are not included.

The results of the mediation analysis showed that emotional responses to a specific stressor that resulted from the COVID-19 pandemic—experienced or expected loss of income—significantly increased the likelihood of avoiding information. The stressor itself had no significant effect on the avoidance of AMR information; instead, the effect was fully mediated by the emotional response.

## 4. Discussion

An increase in the number of crises that humanity faces may generate emotional responses that affect decision-making in a host of important domains. Disease outbreaks and pandemics

**Table 7. Mediation analysis estimating direct and indirect effects of income loss COVID-19 stressor (independent variable) on hopelessness (mediator) and avoidance behaviors (AMR) information avoidance (AMR).**

| | AMR (No demog. variables) | AMR (Demog. variables) |
|---|---|---|
| | Est. (95% Conf. Int.) | Est. (95% Conf. Int.) |
| Average Causal Mediation Effect (Control) | 0.0154*** (0.004, 0.03) | 0.010* (0.001, 0.02) |
| Average Causal Mediation Effect (Treated) | 0.0149*** (0.004, 0.03) | 0.009* (0.0001, 0.02) |
| Average Direct Effect (Control) | -0.021 (-0.072, 0.03) | -0.055 (-0.115, 0.00) |
| Average Direct Effect (Treated) | -0.021 (-0.074, 0.03) | -0.056 (-0.117, 0.00) |
| Total Effect | -0.006 (-0.062, 0.05) | -0.046 (-0.11, 0.01) |
| Proportion Mediated (Control) | -2.532 (-7.145, 6.03) | -0.211 (-1.16, 0.63) |
| Proportion Mediated (Treated) | -2.468 (-7.003, 5.89) | -0.200 (-1.12, 0.64) |
| Average Causal Mediation Effect (Averaged across control/ treated) | 0.0152*** (0.004, 0.03) | 0.009* (0.001, 0.02) |
| Average Direct Effect (Averaged across control/treated) | -0.021 (-0.074, 0.03) | -0.055 (-0.116, 0.00) |
| Proportion Mediated (Average) | -2.500 (-7.074, 5.96) | -0.21 (-1.14, 0.64) |

Notes: *** = <0.001

* = <0.05. Mediation analyses were conducted with 1000 boot-strapped simulations to estimate standard errors.

[62, 66], climate change [67], political polarization [68], and conflicts [69, 70] all evoke strong emotions. The results from our study suggest that these crises may affect decision-making in other domains, which are not directly tied to the crisis.

In this study, we find consistent, significant evidence that the COVID-19 pandemic increased avoidance of information about a distinct but important issue threatening the healthcare system and human health: AMR. Avoidance of AMR information may decrease support for public efforts to address the threat of AMR, such as tightening regulations on the use of antimicrobials or incentivizing development of new antimicrobials, an area of research and development that has stagnated in recent decades [59].

Our analyses suggest that emotion-mediated behavior changes can spread the impact of crises into unrelated domains. Threats such as climate change, conflicts, pandemics, and natural disasters affect behavior directly by changing the costs of actions like going to work or to the doctor's office [70]. However, our findings suggest that—even beyond the fear of contracting COVID-19 during a visit to a healthcare facility—these crises change behavior in ways that are not directly tied to the crisis and that may amplify problems in other important domains. For instance, some of the behaviors leading to the "crisis of undiagnosed cancers" may be due to emotion-mediated decision-making that led individuals to be more risk-averse in their choices about going to a medical facility or that caused them to avoid learning about whether they had cancer [25].

Comparing our AMR survey data with population level data in the US suggests our findings may be a conservative estimate of the impact of the COVID-19 pandemic on decision-making. While respondents to the AMR survey closely matched US population statistics for gender and age, they were markedly more educated, with over 50% of respondents who are 25 years of age or older having completed a four-year college degree, versus approximately 30% for the US population. Individuals with higher levels of education were significantly less likely to experience feelings of hopelessness, which is consistent with previous research [71], potentially yielding a conservative estimate of the link between crises and information avoidance. Further, only 30.1% of the AMR sample reported that they had experienced a decrease in income during the pandemic versus 40.4% of respondents to an overlapping wave of the US Census

Bureau's Household Pulse Survey, suggesting that levels of hopelessness reported in the AMR survey may underestimate one of two focal COVID-19 stressors as well.

Our findings contribute to discussions of policy, poverty, and decision-making. Preliminary evidence suggests that the timing of disbursement of US government support to households in response to the COVID-19 pandemic correlated with improvements in mental health measures [72]. Research on decision-making under conditions of poverty shows that being in a state of poverty impedes cognitive function [73] and changes the allocation of attention [74], which may lead to suboptimal behaviors in a variety of domains, including health, finances, and work [75–77]. Intriguingly, research on poverty and cognition found that cognitive function was only lower when the low-income individual was stressed by a large (hypothetical) unanticipated expense—but not otherwise [73], suggesting that acute stressors alter cognition and decision-making. Our findings that AMR information avoidance increases with emotional reactance to a stressor suggest an additional pathway through which inferior outcomes may be generated—by reducing the information the individual acquires for use in decision-making.

Our study does have some limitations. As noted above, respondents to the AMR survey have significantly higher levels of education on average than the US population, which may influence the generalizability of our findings. However, Ross and Mirowsky [71] find that higher levels of education promote emotional well-being, suggesting that—given the overrepresentation of highly educated individuals—our estimates of the relationship between stressors and hopelessness may be conservative. Our results also show that more highly educated respondents are less likely to report feelings of hopelessness.

The impact of acute crises—such as climate change-related extreme weather events, pandemics, or conflict—has been documented to affect individuals' physical ability to access medical care [70]; impact health directly [70, 78]; have long-term mental health effects, such as post-traumatic stress disorder [79]; and decrease material well-being [80]. Our research suggests that there are also important crisis-driven changes in behavior that may influence decision-making in ways that can threaten future health. Not accounting for these behavioral responses may underestimate the benefits of policies that reduce the impact of these large-scale crises.

## 5. Conclusion

In this article, we examine the relationship between COVID-19-related stressors, hopelessness, and avoidance of information about AMR, which threatens human and animal healthcare systems. Results show strong positive relationships between stressors, hopelessness, and information avoidance behaviors; hopelessness appears to be an important mediator of the effect of stressors on AMR information avoidance. While a decrease in mental health due to COVID-19 has been widely noted, these results suggest relationships between worsening mental health and decision-making in other important areas. Future research should examine broader impacts of emotional responses to crises, particularly in a situation that permits documentation of causality.

## Author Contributions

**Conceptualization:** Christopher R. Gustafson.

**Data curation:** Christopher R. Gustafson.

**Formal analysis:** Christopher R. Gustafson.

**Funding acquisition:** Christopher R. Gustafson, Kathleen R. Brooks, Amalia Yiannaka.

**Investigation:** Christopher R. Gustafson.

**Methodology:** Christopher R. Gustafson, Kathleen R. Brooks, Syed Imran Ali Meerza, Amalia Yiannaka.

**Project administration:** Christopher R. Gustafson, Kathleen R. Brooks, Amalia Yiannaka.

**Resources:** Christopher R. Gustafson.

**Software:** Christopher R. Gustafson.

**Supervision:** Christopher R. Gustafson.

**Validation:** Christopher R. Gustafson.

**Writing – original draft:** Christopher R. Gustafson.

**Writing – review & editing:** Christopher R. Gustafson, Kathleen R. Brooks, Syed Imran Ali Meerza, Amalia Yiannaka.

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
