## [Decision Letter · Decision Letter 0]

23 Aug 2022

PONE-D-22-17318Emotional responses to COVID-19 stressors affect broader health decision-makingPLOS ONE

Dear Dr. Gustafson,

Thank you for submitting your manuscript to PLOS ONE. After careful consideration, we feel that it has merit but does not fully meet PLOS ONE’s publication criteria as it currently stands. Therefore, we invite you to submit a revised version of the manuscript that addresses the points raised during the review process.

We look forward to receiving your revised manuscript.

Kind regards,

Christoph Strumann

Academic Editor

PLOS ONE

Journal Requirements:

2. Please provide additional details regarding participant consent. In the Methods section, please ensure that you have specified (1) whether consent was informed and (2) what type you obtained (for instance, written or verbal). If your study included minors, state whether you obtained consent from parents or guardians. If the need for consent was waived by the ethics committee, please include this information.

Additional Editor Comments:

The manuscript has been evaluated by two reviewers. Both highlighted in their comments that the article needs a revision before it could be accepted for publication. While Reviewer 2 requests a major revision to significantly reduce the length of the manuscript, Reviewer 1’s major concern is the use of the term "causality". Since all individuals have been exposed by the pandemic outbreak, a control group is missing invalidating the use of a natural experiment. The authors should make the related assumptions more explicit, why it is still possible to base the analysis on a natural experiment. Please state explicitly why you are convinced that your findings can be interpreted in a causal manner. The DAG (=directed acyclic graph) figure suggested by Reviewer#1 could be helpful here. Otherwise, Reviewer 1 suggests to avoid any language of causality.

Reviewers' comments:

Reviewer's Responses to Questions

**Comments to the Author**

1. Is the manuscript technically sound, and do the data support the conclusions?

Reviewer #1: Yes

Reviewer #2: Yes

2. Has the statistical analysis been performed appropriately and rigorously? 

Reviewer #1: Yes

Reviewer #2: Yes

3. Have the authors made all data underlying the findings in their manuscript fully available?

Reviewer #1: Yes

Reviewer #2: Yes

4. Is the manuscript presented in an intelligible fashion and written in standard English?

Reviewer #1: Yes

Reviewer #2: Yes

5. Review Comments to the Author

Reviewer #1: The manuscript presents two surveys (custom and a national survey) in which respondents who experienced higher stress in the form of income loss and local case counts felt more hopelessness, and hopelessness was associated with avoiding health information and delaying healthcare.

The manuscript is generally well-written, though some sections repeat what was previously presented. The N is large and being a representative sample makes the finding stronger. I liked the manuscript; I think it adds to the literature. The only major concern is the language of causality in a non-experimental design, but this can be easily remedied.

Some comments:

The title needs to be more specific, addressing avoidance of health information and healthcare utilization.

Under abstract:

Substitute the word "lead" to a non-causal word.

Mental health is more than hopelessness, so indeed be specific and drop 'mental health'.

The fact that there are 2 separate studies is not well-conveyed in the abstract. State explicitly that there are 2 separate studies.

Under introduction.

The statement "Information shapes beliefs and affects private behavior" needs to be supported by other references than #36 and 37, which do not examine this statement.

The authors should explain why they examined accessing treatment for non-COVID health problems as one of the DVs, an already-documented issue. Were they trying to replicate findings? To show the same findings using a big sample? To examine the process/mediators? Please present the rationale.

Structure of writing: LL 162-189 report results, and this is the introduction. The results have no place in the introduction section.

Remove words of causation or prediction from the manuscript. As most measures were taken at same or equivalent times, the word 'predicted'/'predictor' or 'increase' or 'effect' should be replaced with words of association (l. 164, 262, and elsewhere).

Please provide a DAG (=directed acyclic graph) figure conceptualizing the relationship between all variables in the study.

Under methods.

Structure each survey under respondents, materials, procedure, design. The writing is not structured enough.

Explain why a four-category variable of hopelessness made into a 2-category variable.

Streamline the analysis section. The rationale is repeated here after being presented above. There is no need for this.

On l. 312 and 341 remove the word "impacts". No experiment was performed. This is a cross sectional survey and variables have no impact on each other.

2.2.1 – it seems to me the analyses are only on the ARM study, so it's unclear why the title refers also to HPS21. HPS21 is explained under 2.2.2. The national survey appears first under 2.2.1 on l. 333 as an outcome, but this is unclear how it can be an outcome in the costumed survey.

Under Results.

The descriptive data is very interesting, especially the variance between surveys.

Pls substitute 'effect' with association in l. 472. Also impact in l. 488 and 496. The same re 'impact' all over the results.

The authors engage in a lot of interpretation in the results section, and may consider reserving it for the discussion (for example, ll. 503-508).

Table 8 is not very clear in terms of what happens with each stressor; it seems both independent variables were lumped together. A graphic representation could be helpful.

In Table 10, the difference between model I and II, and model III and IV is not clear; one surmises it's the inclusion of demographic variables, but it should be explicitly stated.

Under discussion.

Compare the variables of foregoing medical care to international data, preferably to high resource countries where medical care is less a matter of out-of-pocket payments than in the US. Delayed care was documented in many countries, but the levels may vary. There is such data, and it may increase the claim that the emotional response may drive the decision and behavior and not income issues.

Reviewer #2: Thank you for this thoughtful and relevant article. I appreciate the authors' willingness to step out of their comfort zones of their historically published work and incorporate the relevance of the COVID-19 pandemic with health-related decision making. The authors wove into this work the relevance of AMR and the impact on public health.

This paper is excellently written in outstanding English with appropriate grammar and sentence structure. I am requesting a major revision as it needs significant reduction in length to make this a digestible and relevant read for the reader.

In summary

Introduction:

I would suggest greatly reducing the introduction to 2 pages, to make a very concise and compelling draw for the reader to fully read this work. Please consider utilizing the paragraph starting in line 80 and ending in line 85 ("In this study, we examine a novel question: can one crisis—the COVID-19 pandemic—change decision-making in unrelated but critical personal and societal health issues by affecting general feelings of hopelessness? We use a natural experiment—variation in exposure to COVID-19-related stressors (income loss and local COVID-19 cases)—to examine the impact of a large-scale crisis on decisions to forego or delay necessary medical care and to avoid information about AMR.") to be the final paragraph of the introduction. It is an excellent description of the study. Any relevant information you would like to include, I would suggest placing before this paragraph.

Please avoid the study description in the introduction itself. It creates some confusion and description/methods should be kept entirely to the methods section. Lines 102 through 189 should be greatly condensed to the methods section or omitted if possible.

Methods:

Please attempt to condense the methods as well to 2-3 pages and avoid too much explanation, such as line 206, as the methods should be straight forward and to the point so the reader is not bogged down in the reading prior to reviewing the results section.

Results:

Well-written and contains a wealth of data. It may be beneficial to move some of the data to Supplemental tables as 10 Tables is on the heavy end for a manuscript. The suggestion here is to limit your results to 5 tables and speak to the other data either in writing or by presenting as Supplemental material.

Please avoid sentences like 374-376 which are better suited for the discussion (and a similar sentence was found in the discussion).

Discussion:

This is concise and well-presented. I appreciate the understanding of limitations on the study.

Conclusion:

This was not provided - kindly write a brief 1 paragraph conclusion of the findings and relevance to future work.

Thank you

6. PLOS authors have the option to publish the peer review history of their article (what does this mean?). If published, this will include your full peer review and any attached files.

Reviewer #1: No

Reviewer #2: No

---

## [Author Response · Author response to Decision Letter 0]

23 Sep 2022

The manuscript has been evaluated by two reviewers. Both highlighted in their comments that the article needs a revision before it could be accepted for publication. While Reviewer 2 requests a major revision to significantly reduce the length of the manuscript, Reviewer 1’s major concern is the use of the term "causality". Since all individuals have been exposed by the pandemic outbreak, a control group is missing invalidating the use of a natural experiment. The authors should make the related assumptions more explicit, why it is still possible to base the analysis on a natural experiment. Please state explicitly why you are convinced that your findings can be interpreted in a causal manner. The DAG (=directed acyclic graph) figure suggested by Reviewer#1 could be helpful here. Otherwise, Reviewer 1 suggests to avoid any language of causality.

Response to Editor: We thank the editor for contextualizing the reviewers’ comments. We have addressed each reviewer comment. Language claiming causality has been removed from the manuscript and we added a DAG. We have also made the cuts that the other reviewer requested to make the intro and methods sections markedly shorter and cut down the number of tables to 5. We have moved some text (specifically an expanded materials and methods section) and tables to supplementary materials in case readers would like to refer to them.

Reviewers' comments:

Reviewer's Responses to Questions

Comments to the Author

1. Is the manuscript technically sound, and do the data support the conclusions?

Reviewer #1: Yes

Reviewer #2: Yes

2. Has the statistical analysis been performed appropriately and rigorously? 

Reviewer #1: Yes

Reviewer #2: Yes

3. Have the authors made all data underlying the findings in their manuscript fully available?

Reviewer #1: Yes

Reviewer #2: Yes

4. Is the manuscript presented in an intelligible fashion and written in standard English?

Reviewer #1: Yes

Reviewer #2: Yes

5. Review Comments to the Author

Reviewer #1: 

Comment: The manuscript presents two surveys (custom and a national survey) in which respondents who experienced higher stress in the form of income loss and local case counts felt more hopelessness, and hopelessness was associated with avoiding health information and delaying healthcare.

The manuscript is generally well-written, though some sections repeat what was previously presented. The N is large and being a representative sample makes the finding stronger. I liked the manuscript; I think it adds to the literature. The only major concern is the language of causality in a non-experimental design, but this can be easily remedied.

Response: We thank the reviewer for the careful attention to the manuscript. We have responded to every comment and believe that the suggestions from both reviewers’ have helped improve (and streamline) the manuscript.

Some comments:

Comment: The title needs to be more specific, addressing avoidance of health information and healthcare utilization.

Response: We propose editing the title to “Emotional response to COVID-19 stressors increase avoidance of health information and access to care.”

Under abstract:

Comment: Substitute the word "lead" to a non-causal word.

Response: We have edited the abstract to remove “lead.”

Comment: Mental health is more than hopelessness, so indeed be specific and drop 'mental health'.

Response: We have removed the term mental health from the abstract. Where appropriate, we have retained it in the manuscript when, for instance, citing articles that discuss mental health.

Comment: The fact that there are 2 separate studies is not well-conveyed in the abstract. State explicitly that there are 2 separate studies.

Response: We have added language clarifying that there are two outcomes analyzed. In lines 29-30, the text now reads, “In the analysis of two health behaviors that use data from 1) a custom AMR survey and 2) the U.S. Census’s Household Pulse Survey, which asked about accessing healthcare….”

Under introduction.

Comment: The statement "Information shapes beliefs and affects private behavior" needs to be supported by other references than #36 and 37, which do not examine this statement.

Response: We have added references to two articles that examine private behavior in two different domains: farmers’ behavior in the face of climate change information and mitigation behaviors (Arbuckle Jr, J. G., Morton, L. W., & Hobbs, J. (2015). Understanding farmer perspectives on climate change adaptation and mitigation: The roles of trust in sources of climate information, climate change beliefs, and perceived risk. Environment and behavior, 47(2), 205-234.) and the impact of real-time updating vs. piecemeal (per-item) calorie information on beliefs about total calories ordered and on the number of calories actually ordered in a multi-item prepared food choice environment (Gustafson, C. R., & Zeballos, E. (2019). Cognitive aids and food choice: Real-time calorie counters reduce calories ordered and correct biases in calorie estimates. Appetite, 141, 104320).

Comment: The authors should explain why they examined accessing treatment for non-COVID health problems as one of the DVs, an already-documented issue. Were they trying to replicate findings? To show the same findings using a big sample? To examine the process/mediators? Please present the rationale.

Response: The rationale was to examine the relationship between economic impacts of COVID-19, mediators, and accessing treatment. Text describing this rationale has been added to the manuscript: “While the impact of COVID-19 on medical care has been reported in other studies, we chose to examine decisions to forego or delay medical care for non-COVID health issues and to avoid AMR information because both pose an important threat to human health, like COVID-19, and should therefore have been highly salient to respondents, but both decisions may also be related to hopelessness” (lines 120-125).

Comment: Structure of writing: LL 162-189 report results, and this is the introduction. The results have no place in the introduction section.

Response: The other reviewer requested that the introduction be cut at the paragraph that starts with the line, “In this study, we examine a novel question: can one crisis—the COVID-19 pandemic—change decision-making in unrelated but critical personal and societal health issues by affecting general feelings of hopelessness?” We have moved the text to the appropriate sections.

Comment: Remove words of causation or prediction from the manuscript. As most measures were taken at same or equivalent times, the word 'predicted'/'predictor' or 'increase' or 'effect' should be replaced with words of association (l. 164, 262, and elsewhere).

Response: This text has been cut from the manuscript.

Comment: Please provide a DAG (=directed acyclic graph) figure conceptualizing the relationship between all variables in the study.

Response: We have included a directed acyclic graph in the introduction section (figure 1).

Under methods.

Comment: Structure each survey under respondents, materials, procedure, design. The writing is not structured enough.

Response: In response to the other reviewer’s suggestion to cut the methods to two to three pages, much of the information under methods has been cut. To reduce the text by that amount, detail that would be of interest to some but not all readers has been moved to supplementary materials.

Comment: Explain why a four-category variable of hopelessness made into a 2-category variable.

Response: There was apparently a misunderstanding. The four-category variable of hopelessness was not made into a two-category variable. However, the text covering definition of variables at that level has been moved to supplementary materials rather than in the manuscript in response to the other reviewer’s request to reduce the materials and methods section to three pages or fewer. 

Comment: Streamline the analysis section. The rationale is repeated here after being presented above. There is no need for this.

Response: In response to this reviewer’s and the second reviewer’s comments, the analysis section has been streamlined. 

Comment: On l. 312 and 341 remove the word "impacts". No experiment was performed. This is a cross sectional survey and variables have no impact on each other.

Response: This use of “impacts” has been removed from the manuscript.

Comment: 2.2.1 – it seems to me the analyses are only on the ARM study, so it's unclear why the title refers also to HPS21. HPS21 is explained under 2.2.2. The national survey appears first under 2.2.1 on l. 333 as an outcome, but this is unclear how it can be an outcome in the costumed survey.

Response: This is no longer an issue with the streamlined, significantly shorter methods section.

Under Results.

Comment: The descriptive data is very interesting, especially the variance between surveys.

Response: We agree with the reviewer.

Comment: Pls substitute 'effect' with association in l. 472. Also impact in l. 488 and 496. The same re 'impact' all over the results.

Response: These changes have been made.

Comment: The authors engage in a lot of interpretation in the results section, and may consider reserving it for the discussion (for example, ll. 503-508).

Response: We have removed this text from the results section.

Comment: Table 8 is not very clear in terms of what happens with each stressor; it seems both independent variables were lumped together. A graphic representation could be helpful.

Response: The results in this table were based on the income loss stressor as the independent variable. This has been made clear in the title of the table. 

Comment: In Table 10, the difference between model I and II, and model III and IV is not clear; one surmises it's the inclusion of demographic variables, but it should be explicitly stated.

Response: It was the inclusion of demographic and state/time control variables; this has been clarified in the table heading. The other reviewer requested that the total number of tables in the paper be decreased to five (from 10). This table has now been combined with the analysis of the single-round Household Pulse Survey (week 21) and is in table 5 in the revised manuscript. Other tables have been moved to supplementary materials.

Under discussion.

Comment: Compare the variables of foregoing medical care to international data, preferably to high resource countries where medical care is less a matter of out-of-pocket payments than in the US. Delayed care was documented in many countries, but the levels may vary. There is such data, and it may increase the claim that the emotional response may drive the decision and behavior and not income issues.

Response: We thank the reviewer for this thought and have added relevant literature to the discussion section. However, it is also true that our results do not suggest that income issues are absent from avoidance of healthcare; instead, they suggest that the impact may be larger because income loss leads to an emotional response. The results about AMR illustrate this (since there were no differential costs to access the two videos, income should not have had—and did not have based on estimates—a direct relationship with avoiding AMR information. However, hopelessness (and higher levels of hopelessness) did have a significant relationship to information avoidance.

Reviewer #2: 

Thank you for this thoughtful and relevant article. I appreciate the authors' willingness to step out of their comfort zones of their historically published work and incorporate the relevance of the COVID-19 pandemic with health-related decision making. The authors wove into this work the relevance of AMR and the impact on public health.

This paper is excellently written in outstanding English with appropriate grammar and sentence structure. I am requesting a major revision as it needs significant reduction in length to make this a digestible and relevant read for the reader.

Response: We thank the reviewer for their time and thoughtful analysis of the manuscript. We believe that the manuscript has been improved with the suggestions from both reviewers.

In summary

Introduction:

Comment: I would suggest greatly reducing the introduction to 2 pages, to make a very concise and compelling draw for the reader to fully read this work. Please consider utilizing the paragraph starting in line 80 and ending in line 85 ("In this study, we examine a novel question: can one crisis—the COVID-19 pandemic—change decision-making in unrelated but critical personal and societal health issues by affecting general feelings of hopelessness? We use a natural experiment—variation in exposure to COVID-19-related stressors (income loss and local COVID-19 cases)—to examine the impact of a large-scale crisis on decisions to forego or delay necessary medical care and to avoid information about AMR.") to be the final paragraph of the introduction. It is an excellent description of the study. Any relevant information you would like to include, I would suggest placing before this paragraph.

Response: We appreciate the suggestion by the reviewer. We have made the highlighted paragraph the final paragraph of the introduction; as suggested, we moved some text that we view as important to appear before this paragraph and added some text requested by the other reviewer to this paragraph as well. As a result, the introduction is over two pages (but under three pages) long.

Comment: Please avoid the study description in the introduction itself. It creates some confusion and description/methods should be kept entirely to the methods section. Lines 102 through 189 should be greatly condensed to the methods section or omitted if possible.

Response: We have removed the text highlighted by the reviewer.

Methods: 

Comment: Please attempt to condense the methods as well to 2-3 pages and avoid too much explanation, such as line 206, as the methods should be straight forward and to the point so the reader is not bogged down in the reading prior to reviewing the results section.

Response: The methods section has been reduced to fit into less than three pages. Some details have been removed to online supplementary materials. 

Results: 

Comment: Well-written and contains a wealth of data. It may be beneficial to move some of the data to Supplemental tables as 10 Tables is on the heavy end for a manuscript. The suggestion here is to limit your results to 5 tables and speak to the other data either in writing or by presenting as Supplemental material.

Response: Thank you for the suggestion. We have reduced the number of tables in the manuscript to 5 and moved others to supplemental materials to make them accessible to readers who are interested.

Comment: Please avoid sentences like 374-376 which are better suited for the discussion (and a similar sentence was found in the discussion).

Response: This sentence (and others identified in the results section) have been removed the results section and moved to or combined with existing text in the discussion section. The text noted here, for instance, was added to text in the discussion section in the sentence, “Individuals with higher levels of education were significantly less likely to experience feelings of hopelessness, which is consistent with previous research [46], potentially yielding a conservative estimate of the link between crises and information avoidance.”

Discussion:

Comment: This is concise and well-presented. I appreciate the understanding of limitations on the study.

Response: We thank the reviewer for their careful reading of the manuscript and suggestions that we feel have improved the manuscript.

Conclusion:

Comment: This was not provided - kindly write a brief 1 paragraph conclusion of the findings and relevance to future work.

Response: We have added a conclusions section after the discussion section.

Thank you

Response: We thank the reviewer for their constructive comments.

---

## [Decision Letter · Decision Letter 1]

20 Mar 2023

PONE-D-22-17318R1Emotional responses to COVID-19 stressors increase avoidance of health information and access to carePLOS ONE

Dear Dr. Gustafson,

Thank you for submitting your manuscript to PLOS ONE. After careful consideration, we feel that it has merit but does not fully meet PLOS ONE’s publication criteria as it currently stands. Therefore, we invite you to submit a revised version of the manuscript that addresses the points raised during the review process.

Two reviewers have evaluated the revision of the manuscript. Unfortunately, one of the previous reviewers was unable to review. After a very time-consuming search for a new reviewer, I am now able to make a decision on your manuscript, i.e. the article needs to be revised before it can be accepted for publication. I agree with Reviewer #3 that it would be a cleaner contribution to the literature if the authors focus the manuscript on how one existential threat (COVID) can reduce information seeking about another existential threat (AMR). This would allow you to rewrite the manuscript to be about information avoidance rather than switching between decision-making, avoidance behavior, or information avoidance. It would also give you the space to properly report the mediation analysis, which has not received sufficient attention in the methods or results. In addition, a stronger focus would help to increase the clarity of the study.

We look forward to receiving your revised manuscript.

Kind regards,

Christoph Strumann

Academic Editor

PLOS ONE

Reviewers' comments:

Reviewer's Responses to Questions

**Comments to the Author**

1. If the authors have adequately addressed your comments raised in a previous round of review and you feel that this manuscript is now acceptable for publication, you may indicate that here to bypass the “Comments to the Author” section, enter your conflict of interest statement in the “Confidential to Editor” section, and submit your "Accept" recommendation.

Reviewer #2: All comments have been addressed

Reviewer #3: (No Response)

2. Is the manuscript technically sound, and do the data support the conclusions?

Reviewer #2: Yes

Reviewer #3: No

3. Has the statistical analysis been performed appropriately and rigorously? 

Reviewer #2: Yes

Reviewer #3: I Don't Know

4. Have the authors made all data underlying the findings in their manuscript fully available?

Reviewer #2: Yes

Reviewer #3: Yes

5. Is the manuscript presented in an intelligible fashion and written in standard English?

Reviewer #2: Yes

Reviewer #3: No

6. Review Comments to the Author

Reviewer #2: Dear Author - Thank you so much for the extensive revisions to the manuscript. Your attention to detail in updated is noted and appreciated.

Reviewer #3: I think that the framing of the manuscript as a study of decision-making is too broad. My recommendation would be to narrow the scope to information avoidance and remove the mention of decision making and situating the study in the larger field of decision making (impact of affect on decision-making or financial scarcity and decision making etc…). I don't think we can say with any confidence that the effects described in these other decision making studies apply to information avoidance. The same goes for describing the outcome as health behavior (abstract) which will likely confuse the reader.

Not seeking healthcare is a broader and more multiply determined phenomenon than a single information avoidance decision about viewing a video about AMR. I think that lumping them together does a disservice to both. My preference would be to see the two studies split apart into two separate papers and the intro and discussions reworked to focus on the different outcome variables. That said, several of my comments assume that this will not be the case.

The most relevant literature reviews seem underdeveloped. I expected to read a more extensive summary of (1) studies that demonstrated an increase in hopelessness and other depressive symptoms during COVID, (2) determinants of information avoidance and, (3) the impact of COVID on health care seeking and justifying why we can consider this information avoidance or even avoidant behavior. The later is a real sticking point for me. We don’t have information on why people didn’t have care. Perhaps there was no care to receive. That was an issue for many during the pandemic. I appreciate that hopelessness predicted not receiving care so it seems like the pathway is similar to the AMR study, but I didn’t find enough information about the mediation model to feel reassured that there was true causal mediation taking place.

I realize that a previous reviewer requested a shorter methods section and all the measures descriptions have been removed. This seems unusual but I defer to the editor. If they are added back in they could be rewritten to be clearer. I would give each construct its own mini paragraph and describe question wording and response options.

You might consider switching the order of paragraphs 1 and 2 on page 7 so you introduce the study design first and follow that with what people did in the survey. One thing that might help with clarity would be to describe what the participants did in the studies rather than the survey doing things to people.

I don't think the analysis section indicates weighting/approach to using complex survey data unless that's in supplemental materials.

Lines 166-167 in the methods describe 4 nested models. I believe various covariates were added to the model. Please clarify this section so that the reader understand which variables were included in which models and why. These methods don't need to be reiterated in the results but I think a summary of the results and any important differences between the models could be articulated more clearly.

It’s unusual for the results to be written in present rather than past tense.

While some descriptive stats for the two samples makes sense, I didn’t understand the logic of presenting these as a comparison, especially since they aren’t compared in any inferential way.

The stats are presented in various ways in the narrative OR and 95% CI, just the OR or just an approximation of the odd (e.g., 1.5 times). Could this be standardized so that the OR, 95% CI and p-value are always presented?

Could asterisks indicating p-values be added to the tables?

In the interest of preciseness the results only need to be reported once. For example in the following the first sentence can be dropped although the stats for COVID cases seem to be missing so that should be added. I’d do a careful edit for redundancy. " In the analysis of HPS21 data, experienced loss of income, expected loss of income, and state-level COVID-19 cases increase the odds of experiencing hopelessness. Experienced income loss (aOR: 2.42, 95% CI: 2.35, 3.50) and expected income loss (aOR: 2.75, 95% CI: 2.49, 3.04) significantly increase the likelihood that individuals experience higher levels of hopelessness.”

Lines 255-260 seem to editorialize on the results and would seem better apportioned into parts that belong in the methods and the limitations.

7. PLOS authors have the option to publish the peer review history of their article (what does this mean?). If published, this will include your full peer review and any attached files.

Reviewer #2: No

Reviewer #3: No

---

## [Author Response · Author response to Decision Letter 1]

10 May 2023

Reviewer #3: I think that the framing of the manuscript as a study of decision-making is too broad. My recommendation would be to narrow the scope to information avoidance and remove the mention of decision making and situating the study in the larger field of decision making (impact of affect on decision-making or financial scarcity and decision making etc…). I don't think we can say with any confidence that the effects described in these other decision-making studies apply to information avoidance. The same goes for describing the outcome as health behavior (abstract) which will likely confuse the reader.

Not seeking healthcare is a broader and more multiply determined phenomenon than a single information avoidance decision about viewing a video about AMR. I think that lumping them together does a disservice to both. My preference would be to see the two studies split apart into two separate papers and the intro and discussions reworked to focus on the different outcome variables. That said, several of my comments assume that this will not be the case.

Response: We have removed all the material about healthcare decisions and retain only the text about the survey on antimicrobial resistance (AMR) and the decision to view or not to view information about AMR.

The most relevant literature reviews seem underdeveloped. I expected to read a more extensive summary of (1) studies that demonstrated an increase in hopelessness and other depressive symptoms during COVID, (2) determinants of information avoidance and, (3) the impact of COVID on health care seeking and justifying why we can consider this information avoidance or even avoidant behavior. The later is a real sticking point for me. We don’t have information on why people didn’t have care. Perhaps there was no care to receive. That was an issue for many during the pandemic. I appreciate that hopelessness predicted not receiving care so it seems like the pathway is similar to the AMR study, but I didn’t find enough information about the mediation model to feel reassured that there was true causal mediation taking place.

Response: The literature in point 3) is no longer relevant as the paper has been refocused to consider only AMR information avoidance. We have increased the text devoted to emotional responses to COVID, with a particular focus on hopelessness (lines 52-73). We have also added to the literature on information avoidance (lines 85-108). Additions to the text are highlighted in yellow.

I realize that a previous reviewer requested a shorter methods section and all the measures descriptions have been removed. This seems unusual but I defer to the editor. If they are added back in they could be rewritten to be clearer. I would give each construct its own mini paragraph and describe question wording and response options.

Response: We have re-incorporated text that had been moved out of the paper, and retained only the information about the AMR study. This section is markedly different and changes have been highlighted in yellow.

You might consider switching the order of paragraphs 1 and 2 on page 7 so you introduce the study design first and follow that with what people did in the survey. One thing that might help with clarity would be to describe what the participants did in the studies rather than the survey doing things to people.

Response: We have changed the order of the paragraphs based on the suggestion of the reviewer. Further, we have additionally attempted to change the language so that it describes what the participants did.

I don't think the analysis section indicates weighting/approach to using complex survey data unless that's in supplemental materials.

Response: We have removed the HPS data from the paper.

Lines 166-167 in the methods describe 4 nested models. I believe various covariates were added to the model. Please clarify this section so that the reader understand which variables were included in which models and why. These methods don't need to be reiterated in the results but I think a summary of the results and any important differences between the models could be articulated more clearly.

Response: We have added text to clarify the variations in the models. The text (lines 242-252) now reads, “Next, we examined the relationship of hopelessness to AMR information avoidance using binary logistic regression models. The outcome variable documenting AMR information avoidance was the choice to access or avoid a video about AMR in food and agriculture, which is a binary variable (=1 if the respondent chose to avoid the video about food, agriculture, and AMR; and =0 if they chose to access the AMR video). A second version added local cases of COVID-19 and income loss in addition to hopelessness to identify whether these COVID-19-related stressors directly influence AMR information avoidance behavior or only act indirectly through their effect on hopelessness. Next, we conducted a version of the regression that incorporated demographic variables. Finally, we examined AMR information avoidance behavior with hopelessness, COVID-19 stressors, and demographic characteristics as independent variables.”

It’s unusual for the results to be written in present rather than past tense.

Response: We have changed the results to be written in the past tense.

While some descriptive stats for the two samples makes sense, I didn’t understand the logic of presenting these as a comparison, especially since they aren’t compared in any inferential way.

Response: While the HPS analysis has been removed, we have included descriptive statistics from one wave of the HPS survey that overlapped with the data collection period for the AMR survey as the best available comparison for income loss and hopelessness from a larger US survey at a similar point in time. These data are reported in tables 2 and 4. 

The stats are presented in various ways in the narrative OR and 95% CI, just the OR or just an approximation of the odd (e.g., 1.5 times). Could this be standardized so that the OR, 95% CI and p-value are always presented?

Response: We have adopted the reviewer’s requested format for reporting.

Could asterisks indicating p-values be added to the tables?

Response: We have added asterisks indicating p-values to the tables.

In the interest of preciseness the results only need to be reported once. For example in the following the first sentence can be dropped although the stats for COVID cases seem to be missing so that should be added. I’d do a careful edit for redundancy. " In the analysis of HPS21 data, experienced loss of income, expected loss of income, and state-level COVID-19 cases increase the odds of experiencing hopelessness. Experienced income loss (aOR: 2.42, 95% CI: 2.35, 3.50) and expected income loss (aOR: 2.75, 95% CI: 2.49, 3.04) significantly increase the likelihood that individuals experience higher levels of hopelessness.”

Response: We have gone through the manuscript to try to eliminate any redundancy.

Lines 255-260 seem to editorialize on the results and would seem better apportioned into parts that belong in the methods and the limitations.

Response: This text has been eliminated from the manuscript, as it was about results from the analysis of HPS data.

---

## [Editor Report · Decision Letter 2]

23 May 2023

Emotional responses to COVID-19 stressors increase information avoidance about an important unrelated health threat

PONE-D-22-17318R2

Dear Dr. Gustafson,

We’re pleased to inform you that your manuscript has been judged scientifically suitable for publication and will be formally accepted for publication once it meets all outstanding technical requirements.

Kind regards,

Christoph Strumann

Academic Editor

PLOS ONE
---

## [Editor Report · Acceptance letter]

25 May 2023

PONE-D-22-17318R2 

Emotional responses to COVID-19 stressors increase information avoidance about an important unrelated health threat 

Dear Dr. Gustafson:

I'm pleased to inform you that your manuscript has been deemed suitable for publication in PLOS ONE. Congratulations! Your manuscript is now with our production department. 

Kind regards, 

on behalf of

Dr. Christoph Strumann 

Academic Editor

PLOS ONE